# Human Umbilical Cord Lining-Derived Epithelial Cells: A Potential Source of Non-Native Epithelial Cells That Accelerate Healing in a Porcine Cutaneous Wound Model

**DOI:** 10.3390/ijms23168918

**Published:** 2022-08-10

**Authors:** Jonah Ee Hsiang Kua, Chun Wei Siow, Wee Keng Lim, Jeyakumar Masilamani, Monica Suryana Tjin, Joe Yeong, Tony Kiat Hon Lim, Toan Thang Phan, Alvin Wen Choong Chua

**Affiliations:** 1Department of Plastic, Reconstructive and Aesthetic Surgery, Singapore General Hospital, 20 College Road, Singapore 169856, Singapore; 2Skin Bank Unit, Singapore General Hospital, Outram Road, Singapore 169608, Singapore; 3CellResearch Corporation Pte. Ltd., 137 Market Street, Grace Global Raffles #08-02, Singapore 048943, Singapore; 4Programme in Cardiovascular and Metabolic Disorders, Duke-NUS Medical School, 8 College Road, Singapore 169857, Singapore; 5Institute of Molecular and Cell Biology, Agency for Science, Technology and Research, 61 Biopolis, Proteos, Singapore 138673, Singapore; 6Department of Anatomical Pathology, Singapore General Hospital, 20 College Road, Singapore 169856, Singapore; 7Department of Surgery, Yong Loo Lin School of Medicine, National University of Singapore, Kent Ridge Road, Singapore 119228, Singapore; 8Musculoskeletal Sciences Academic Clinical Programme, Duke-NUS Medical School, 8 College Road, Singapore 169857, Singapore

**Keywords:** umbilical cord, cord lining epithelial cells, skin keratinocytes, cutaneous wounds, wound healing, CLECs

## Abstract

Human umbilical cord lining epithelial cells [CLECs) are naïve in nature and can be ethically recovered from cords that are routinely discarded. The success of using oral mucosal epithelial cells for cornea defects hints at the feasibility of treating cutaneous wounds using non-native CLECs. Herein, we characterized CLECs using flow cytometry (FC) and skin organotypic cultures in direct comparison with skin keratinocytes (KCs). This was followed by wound healing study to compare the effects of CLEC application and the traditional use of human skin allografts (HSGs) in a porcine wound model. While CLECs were found to express all the epidermal cell markers probed, the major difference between CLECs and KCs lies in the level of expression (in FC analysis) as well as in the location of expression (of the epithelium in organotypic cultures) of some of the basal cell markers probed. On the pig wounds, CLEC application promoted accelerated healing with no adverse reaction compared to HSG use. Though CLECs, like HSGs, elicited high levels of local and systemic immune responses in the animals during the first week, these effects were tapered off more quickly in the CLEC-treated group. Overall, the in vivo porcine data point to the potential of CLECs as a non-native and safe source of cells to treat cutaneous wounds.

## 1. Introduction

The success of autologous skin keratinocyte (KC) cultures that have been genetically modified to treat junctional epidermolysis bullosa (JEB) brought the spotlight back on cultured epithelial cell therapy in the field of regenerative medicine [1]. In 2017, De Luca and his team regenerated an entire human epidermis that is fully functional on a 7-year-old child whose devastating form of JEB affected 80% of his total body surface area. However, resurfacing critically large cutaneous wounds due to severe burns, diabetes, or genetic mutations remains a huge clinical challenge. In the current regenerative medicine landscape, one area that could be addressed is to identify an alternative source of epithelial cells that can be used effectively to treat extensive skin barrier defects in the clinics. There are known instances where suitable skin donor sites were not available for culture autologous KCs in life-saving procedures. In regions of tropical climate, high incidences of bacterial infection in severe and extensively burned patients have been reported [2,3,4,5] with limited donor sites available that were already highly colonized for any meaningful skin culture. Similarly, it is known that KCs isolated from mutated skin donor sites of EB patients are challenging to cultivate due to structural and functional abnormalities resulting in a lower growth rate compared to normal KCs [6,7]. Therefore, there is a need to look at non-native or alternative sources of epithelial cells with barrier function to resurface cutaneous wounds, especially those with large surface areas.

There is an emerging trend of using umbilical cord tissues to derive epithelial cells as a cost-effective option in regenerative medicine [8,9]. On top of having the advantages of a naive status and an abundant resource as medical waste, umbilical cords are collected without the need for any deliberate invasive procedures and with minimal ethical considerations [10,11]. The umbilical cord epithelium is known to display certain cytokeratin expressions similar to that of the human epidermis [12], and derived human umbilical cord lining epithelial cells (CLECs) were found to have common features with neonatal epidermal KCs at the morphological and molecular level [13]. In addition, CLECs were not only found to have a low immunogenicity and immunosuppressive function in vitro [14], these cells could also be maintained for extended periods in vivo as compared to KCs [8].

Interestingly, there is no known report to date on the safety, efficacy, and immunogenicity of CLECs used in cutaneous wound healing applications in vivo. Herein, we validated the phenotype of CLECs via flow cytometry and studied their behavior using ex vivo skin organotypic cultures in comparison to human skin KCs. This was followed by topical application of human CLECs on full thickness excisional wounds of immunocompetent pigs to study their wound healing efficacy and immunological effects.

## 2. Results

### 2.1. Immunophenotypic Profile of CLECs in Comparison to KCs

Analysis of the flow cytometry (FC) data found that isolated CLECs displayed some of the immunophenotypic profiles found in KCs with high expression levels of epidermal basal/adhesion cell markers of keratin 5, 14(KRT5, KRT14), and integrin alpha-6 (ITGA6). There were corresponding low levels of differentiated cell markers of keratin 1, 10(KRT1, KRT10), and involucrin (IVL) across all the three passages tested (Figure 1a–c, Appendix A). While more than 50% of the overall CLECs expressed another set of basal/adhesion cell markers of KRT15 (Mean = 55.3%, SD = 4.64%, n = 18) and integrin beta-1 (ITGB1) (Mean = 60.0%, SD = 8.75%, n = 18) across the three passages tested, these numbers were significantly lower than that expressed by KCs for KRT15 (mean = 77.5%, SD = 8.90%, n = 18) and ITGB1 (mean = 80.8%, SD = 7.8%, n = 18).

### 2.2. Stratification of CLECs Compared to KCs in Organotypic Cultures

In organotypic cultures, CLECs were able to form stratified epithelium on skin fibroblast populated de-epithelialized dermis (DEDs), even though they were not as organized and mature as those formed by KCs after two weeks of air-liquid interface exposure (Figure 2a). This was reflected in the immunofluorescence (IF) staining data where basal markers of KRT14, KRT15, and p63 (Figure 2b) were found positive across the basal and suprabasal regions of CLEC-generated epithelia, while the same markers were restricted mainly at the basal area of KC-generated epithelia.

However, differentiated cell markers of KRT10 and IVL (at the suprabasal region) as well as adhesion molecule markers of ITGA6 and ITGB1 (at the basement membrane) region were expressed in the CLEC-generated epithelia, similar to the marker profile/location of the KC-generated epithelia (Figure 2b,c). A further breakdown on the location and distribution of the various IF stained markers expressed within the entire epithelial layer of organotypic cultures formed by both CLECs and KCs were compared side by side in Table 1.

### 2.3. Wound Healing Performance of CLECs in Comparison to Human Skin Grafts (HSGs) in a Porcine Excisional Wound Model

A total of eight pigs, matched for sex, age, and weight, were used for the above study. These large animals, each created with six full-thickness excisional wounds, were subjected to topical treatment of human CLECs as a fresh cell pellet layer (concentration at 10^5^ cells/cm^2^), cryopreserved human skin grafts (HSGs), and non-biological standard dressings (“untreated” control) for comparison. The various configurations of these wound treatments (Figure 3) are described in the Materials and Methods, Section 4.6 (Figure 4). In configuration 1 with a total of four wounds (n = 4) for each treatment arm that were performed across two pigs (Figure 4a), we found accelerated wound closure in the CLEC-treated group that was near significance (*p* = 0.0518) based on repeated measures two-way ANOVA over time (Figure 3a). Between Weeks 2 to 4, the percentage area of wound closure was found to be significantly higher with CLEC treatment compared to HSG and/or standard wound dressing treatment (“Untreated”).

In configuration 2—Set A (Figure 3b and Figure 4b), each of the three pigs were subjected to a single type of treatment and compared across these pigs over a period of 7 weeks. The wounds (n = 5) of the CLEC-treated pig achieved the highest rate of healing, with 99% closure compared to 83% with the HSG-treated pig and 78% for the “untreated” pig just before sacrifice. The CLEC treatment provided an accelerated wound closure that was highly significant (*p* < 0.0001) over time compared to the other two treatment arms.

In configuration 2—Set B (Figure 3c and Figure 4b), where the animals were sacrificed once the average closure of all wounds on the pig achieved 90% or above, we similarly found that the pig that received CLECs performed the best overall. The CLEC-treated pig achieved 98% wound closure (n = 5) at Week 4, while the “untreated” and HSG-treated pigs both achieved 94% wound closure at Week 6 and Week 8, respectively.

Throughout the entire wound inspection period, we observed no obvious signs of infection or severe adverse reaction to the animals in the CLEC and the “untreated groups, except for some swellings and sloughs that were found on the HSG-treated wound beds at Week 2 (Figure 3b,c).

### 2.4. Histological Assessment of the Wounds after Respective Treatments

In Masson’s Trichome (MT) staining of representative wound tissue biopsies, we found the consistent presence of collagen (stained turquoise) with more mature and thicker epithelia formed on the wounds of the two CLEC-treated pigs: one at Week 6 in Configuration 2 Set A and the other at Week 3 in Set B. All of these time points were at one week before the animals were sacrificed (Figure 5a). Further analysis of the MT stains at a higher magnification (Figure 5b) using semi-quantitative scoring method [15] (Table 2) revealed that CLEC-treated skin stained with MT demonstrated strong blue staining indicating new collagen formation in the region of reticular dermis as early as at Week 3. In the Set B experiment, the blue staining could be found not only in the region of reticular dermis but also in the papillary dermis as early as at Week 3, whereas the HSG-treated skin as well as the untreated skin could only demonstrated the similar positive staining pattern at Week 6.

Blinded assessment of immunohistochemistry (IHC) stains of representative wound tissue biopsies (Figure 6) to determine the inflammatory infiltration levels of CD4+ and CD8+ T lymphocytes [16] into the respective wound beds (treated by the three different treatments over time) were all compiled in Table 3. Differences in the CD4 and CD8 ratings between the two pigs treated with the same modality (comparing between Sets A and B) were observed and these could be attributed to pig-to-pig variation in the local immunological response of the wound bed. However, the overall trend seemed to suggest that while CLEC treatment elicited high levels of CD4+ T lymphocyte infiltration at the first week in the two pigs that were treated, and these levels subsequently subsided. This was unlike HSG treatment in which high levels of CD4+ T lymphocyte infiltration were still observable from Week 4/5 onward.

### 2.5. Evaluation of Systemic Host Response through Time-Course Profiling of Probed Inflammatory Cytokines in Porcine Serum with Respective Wound Treatments

The profiling of pro- and anti-inflammatory cytokines present in the blood serum of the differently treated pigs (in Configuration 2 Set A where all pigs were tracked over a period of 7 weeks) revealed that there was an increase in the levels of IFN-γ, IL-8, and TNF-α, all pro-inflammatory cytokines of the CLEC-treated pig on Day 4, relative to the pre-procedure baseline level. The high level of IL-8 at this time point was especially pronounced with a fold increase in the range of 61 to 120. However, this observed increase in the IL-8 level completely subsided from Weeks 1 to 3, only to see its level rise again (fold increase of 31 to 60) on Week 4 (Table 4).

Similarly, the serum of an HSG-treated pig had a pronounced increase in the level of IL-8 on Day 4, which receded totally between Weeks 1 and 2, only to return at elevated levels from Weeks 3 to 6. There was also a substantial increase in the levels of TNF-α that were sustained from Weeks 3 to 6. Furthermore, the use of HSG also brought about sporadic increases in levels of other inflammatory cytokines—IFN-γ, IL-1β, IL-2, IL-6, Il-10, IL-12, and IL-18—at various time points, with most of them concentrated at Week 2. Finally, we found an increase in the level of only one anti-inflammatory cytokine, IL-10, in the pig serum that received standard dressings (“Untreated”) on Weeks 3, 5, and 7 (Table 4).

## 3. Discussion

The concept of using a “non-native” source of cultured epithelial cells for healing and regeneration of different epithelial tissue type is not new. In 2004, the first-in-man corneal reconstruction with cultured cell sheets composed of autologous oral mucosal epithelium was described [17], and this technique is still being used in the clinic today [18]. Drawing on the same principle, we explored the possibility of using human CLECs as an alternative source to human KCs for the treatment of cutaneous wound injuries using Yorkshire pigs. This large-animal excisional wound model has been reported to closely resemble normal healing in humans with neither having excessive contraction nor forming hypercontracted scars [19].

### 3.1. Expression Levels of KRT15 and ITGB1 Were Found to Be Lower in CLECs Compared to KCs in Flow Cytometry

We first characterized CLECs by studying their phenotype in direct comparison to KCs via FC analysis. Isolated CLECs were found to have some immunophenotypical profiles similar to KCs (across all three passages) with high expression levels of epidermal basal keratin markers, KRT5 and KRT14, and the adhesion molecule marker, ITGA6, with correspondingly low levels of differentiated skin epithelial cell markers: KRT1, KRT10, and IVL. However, the expression levels of KRT15 and ITGB1 in CLECs were approximately 20% lower than that found in KCs (Figure 1a–c). The CLEC expression level for ITGB1 of 60% on average, differs from an earlier study, which reported higher expression levels at more than 90% [20]. As that study did not perform a direct comparison with KCs such as ours, the difference in ITGB1 expression levels in CLECs might be attributed to lab-to-lab variation in terms of protocols and reagent use, as well as biological variation from different donors.

### 3.2. Some Basal Cells Markers of the Human Epidermis Were Found to Be Expressed in Both the Basal and the Suprabasal Layers of CLEC-Generated Epithelium

In organotypic cultures, we found that seeded CLECs were able to form a fully stratified epithelium after 2 weeks of air-liquid interface exposure. IF staining showed that CLECs can generate an epithelium expressing basal-(KRT14, KRT15, p63, ITGA6, and ITGB1), suprabasal-(KRT10), and even upper suprabasal-(IVL) cell markers which are all typically found on the human skin epidermis (Figure 2b,c). While CLECs could form a stratified stratum corneum layer with the presence of IVL expression, known for its barrier function, the overall appearance of CLEC-generated epithelium appeared less mature and organized compared to that formed by KCs, as seen on the H&E stains (Figure 2a). Furthermore, differences were detected in the location within the layer of epithelia where these markers were expressed in the respective IF stains of CLEC- and KC-organotypic cultures (Table 1). Essentially, KRT14, KRT15, and p63, known to be expressed in the basal KCs of the skin epidermis [21,22], were found to be more prevalent in the entire epithelium (up to the suprabasal layer) formed by CLECs. This observation correlates with a previous comparative study of umbilical cord and fetal skin tissues, where KRT14 expression was found in the whole epithelium of the cord tissue, while this expression was limited to the basal layer of the fetal skin epidermis [12]. Taken together, we postulate that the differentiation of seeded CLECs might not be as efficient as KCs during the 2-week stratification process. CLECs, being a non-native source cultured on skin dermis, might still be in their basal or naïve state in transit within the suprabasal region. These cells in transition might need more time for full maturation during the stratification process. Future studies and characterization would be needed to study this longer-term stratification process.

### 3.3. CLEC Treatment Accelerated Wound Closure Compared to HSG Treatment

Cutaneous wound healing in the pig is frequently used as a model for human cutaneous wound healing due to the anatomical and physiological similarities between the skin of these two species [19,23]. In our investigation to find out the suitability of human CLECs for cutaneous wound healing application, we tested these cells as a form of xenograft treatment [24] on a porcine full thickness excisional wound model and compared it with the use of cryopreserved human skin grafts (HSGs). Allogeneic HSGs are considered the off-the-shelf gold standard to treat severe wounds in extensive burns after wound excision [25]. In both configurations 1 and 2 (Figure 4), we found general improvement in percentage of wound closure in the pigs treated with CLECs (Figure 3a–c), and this remains so after discounting the effect of wound contraction, which was found to be on average close to 50% for all of the wounds regardless of treatment type at the point of sacrifice (Appendix A). MT stains of collected tissue biopsies corroborate the observed improved wound healing in the CLEC-only-treated pigs with the presence of collagen in both Sets A and B, as well as having more mature and thicker epithelia one week before the animals were sacrificed (Figure 5a and Figure 6). HSG-treated skin showed some positive MT staining only at Week 6, which demonstrated slower wound healing compared to the other groups. We speculate that the immunological response to the human xenograft might be one of the possible reasons.

The improvement in wound healing elicited by CLECs in configuration 1 was not as pronounced in the two pigs treated entirely by CLECs in configuration 2. Here, we hypothesized that there might be a confounding effect of the systemic response on wound healing performance over time due to multiple treatments administered onto a single pig used in configuration 1. And indeed, based on the pigs’ serum evaluation, there were significant differences in the overall profile of the systemic inflammatory cytokines detected over time between the three treatment groups. The treatment arm that brought about the most immunogenic reaction might potentially affect the other more passive treatment arms (Table 4) on the same animal.

### 3.4. Use of CLECs and HSGs Elicited High Levels of Local and Systemic Immune Response in the Animals during the First Week but These Effects Tapered off More Quickly in the CLEC-Treated Group

Blinded IHC assessment of local inflammatory infiltrations of T lymphocytes in the wound biopsies to study immune rejection mediated by these cells [16] showed that treatment of both CLECs and HSGs brought high levels of CD4+ T cells into the wounds in the first week (Table 2). However, the CD4+ T cell distribution in the CLEC-treated group seemed to taper more over time compared to the HSG-treated group. Similarly, we also observed increased levels of pro-inflammatory cytokines detected in the pigs’ serum at Day 4 of the CLEC- and HSG-treated group only to find that these were subsequently muted for the CLEC-treated group in the remaining weeks, save for the detection of IL-8 in Week 4. While we cannot fully explain the moderate presence of CD8+ T cells in almost all weeks of the different treatment arms, we can only postulate that these cells recruited into the wounds have reached a threshold regardless of the trigger that was due to the wounding itself, the immunogenic response to the foreign cells/tissues or a combination of both, and these CD8+ T cell levels were sustained during the entire wound healing process until the wounds were almost or completely epithelialized, quite similar to what was previously reported in mice wound healing [26].

The period of tapering in both the local and systemic immune response of CLEC-treated animals seemed to correspond to the period of improved wound healing compared to the HSG-treated animals. While it is not clear what causes this tapering effect, it might be attributed to the immunoregulatory effects mediated by CLECs as previously reported [8], or it might be due to the difference in the duration of CLECs and HSGs (along with their residues) that persisted on their respective wound bed after topical application. Based on clinical observation and assessment, we detected neither any adverse reaction nor overt immune rejection in animals of the CLEC and “untreated” groups. Swellings and sloughs were found only on the HSG-treated wound beds at Week 2.

## 4. Materials and Methods

### 4.1. Isolation and Culture of Human Cord Lining Epithelial Cells (CLECs)

Four lines of human CLECs were provided by CellResearch Corp, Singapore (see information in Appendix A). These cells were isolated from the lining membrane of human umbilical cords according to previously described protocols [8]. Briefly, human umbilical cord tissues were obtained after delivery of uncomplicated pregnancies with written informed consent from healthy donors based on the ethics approval given by the National University of Singapore. Dissection of the umbilical cord tissue was performed to separate the umbilical cord membrane from Wharton’s jelly and other components. Sectioned small pieces of umbilical cord membrane were explanted on tissue culture dishes and the primary epithelial cells were allowed to grow out of these sections. Cultures of these CLECs were maintained in PTT-e3 medium (CellResearch Corp, Singapore) that is made up of medium 171 (Cascade Biologics, Portland, OR, USA) supplemented 2.5% fetal bovine serum (FBS), 50 μg/mL insulin-like growth factor-1 (IGF-1), 50 μg/mL platelet-derived growth factor-BB (PDGF-BB), 5 μg/mL transforming growth factor-β1 (TGF-β1), and 5 μg/mL insulin. CLECs from passages 2 to 6 were used for downstream in vitro and in vivo experiments.

### 4.2. Isolation and Culture of Human Skin Keratinocytes (KCs)

Four lines of human KCs (see information in Appendix A) were isolated from the waste foreskin of healthy subjects that were donated with informed consent and approved by the ethics review board of SingHealth. Briefly, the collected skin tissue was washed in phosphate-buffered saline (PBS, Lonza, Walkersville, MD, USA) and incubated in 10 mL of 2.5 mgmL^−1^ Dispase II (Roche, Mannheim, Germany) in Dulbecco’s modified Eagle medium (DMEM, Gibco, Thermo Fisher Scientific, Grand Island, NY, USA) and left overnight at 4 °C. The following day, epidermis was mechanically separated from dermis with fine forceps and incubated in 0.05% trypsin-EDTA solution (Gibco) for 15 min at 37 °C. Upon cellular dissociation, trypsin activity was reduced by diluting the solution with three volumes of fresh DMEM. KCs were then collected through centrifugation and resuspended in KGM-CD (Lonza) for downstream culture or cryopreservation. KCs were seeded at a density of 9 × 10^4^ cells/cm^2^ on 6-well plates (Corning, Sigma-Aldrich, New York, NY, USA) pre-coated with laminin 511 [27] and cultured in KGM-CD at 37 °C/7.5% CO_2_ for further expansion. Cells between passages 2 to 6 were used for the in vitro experiments in this study.

### 4.3. Flow Cytometry (FC) and Analysis of Data

Isolated CLECs and KCs were collected from passages 3 to 5. Single-cell suspensions were fixed with Fixation Reagent (Medium A; Life Technologies, Thermo Fisher Scientific, Frederick, MD, USA) for 15 min at room temperature (RT), washed with FC buffer (0.5% bovine serum albumin and 2 mM EDTA in 1× PBS), blocked with 5% goat serum in FC buffer and then immunostained with primary antibodies in Permeabilization Reagent (Medium B; Life Technologies) for 15 min at RT. Primary antibodies and their used dilution factor are shown in Appendix A. This would be followed by detection with secondary antibodies diluted with 1% goat serum in FACS buffer. Secondary antibodies used included rabbit anti-mouse FITC (1:1000, Dako, Glostrup, Denmark), Alexa Fluor 647-conjugated goat anti-rabbit (1:1000, Life Technologies, Carlsbad, CA, USA), and Alexa Fluor 647-conjugated goat anti-mouse (1:1000, Life Technologies). For fluorophore-conjugated antibodies, fixed cells were incubated with antibodies diluted in Medium B and human FcR blocking reagent (1:50, Miltenyi Biotec, Bergisch Gladbach, Germany) for 30 min at RT. Stained cells were resuspended in FC buffer and subjected to analysis (MACSQuant VYB, Miltenyi Biotec) with two technical duplicates for each cell type. FC data were analyzed using MACSQuantify (Miltenyi Biotec) software based on gated live cells.

### 4.4. Reconstruction of Epithelium on De-Epithelialized Dermis (Organotypic Culture)

The epidermis of glycerol-preserved split thickness allogeneic skin tissue (EURO SKIN BANK, EA Beverwijk, Netherland) was mechanically removed after several cycles of snap-freezing in liquid nitrogen and thawing. The remaining de-epithelialized dermis (DED) was cut into 2 × 2 cm squares and with the reticular side of the dermis facing up placed on a 6-well 0.4 μm pore size transparent PET membrane insert (Falcon, Corning, Durham, NC, USA). This insert was then placed within one well of a 6-deep well plate (Corning) and the dermal skin seeded with 5 × 10^5^ human dermal fibroblasts (see information in Appendix A) in 50 μL of DMEM with 10% fetal bovine serum (FBS, Hyclone, Logan, UT, USA). After 30 min, the entire fibroblast-seeded skin was cultured submerged in DMEM with 10% FBS.

The following day, the DED was flipped over and 2 × 10^5^ of fresh KCs or CLECs (in 50μL of medium) was seeded separately on individual DED. After 30 min, the entire organotypic culture was cultured in submerged medium for 7 days in complete F12 and DMEM (cFAD) medium that were changed every 2–3 days. Subsequently, these cultures were lifted to an air–liquid interface to stratify in the same cFAD medium for another 14 days as previously described [28]. Each sample was then processed and cryosectioned for both H&E and immunostaining.

### 4.5. Porcine Full Thickness Excisional Wound Model

Healthy, stress-free Yorkshire-Landrace (Sus scrofa) pigs matched for age (12 weeks old), weight, and sex (female) were obtained from the National Large Animal Research Facility, Singapore. The animal experiments were executed in accordance with the National Advisory Committee for Laboratory Animal Research. These immunocompetent pigs were intramuscularly pre-medicated with ketamine/diazepam and anesthesia maintained with 2% isoflurane. After removal of the dorsal hair at the thoracic region with hair clippers, the exposed skin region was cleaned with chlorhexidine and 70% isopropyl alcohol solution. A template was used to define a total of six wound sites—three bilateral wounds on each side of the back spine on the thoracic region. Each wound measured 5 cm × 5 cm was first tattooed followed by use of no.15 blade to excise each of the marked area to create a full thickness wound (Appendix A).

Two configurations of treatment (Figure 4) were used to test the efficacy of fresh human CLECs applied topically to the six full thickness wounds created on each pig. The number of pigs used were worked out based on the 3 Rs principle in animal experimentation [29].

Configuration 1: As a form of screening, a total of two pigs with each of them subjected to three parallel arms of treatment were compared for wound healing performance over a period of 7 weeks (Figure 4a). The three treatment arms were: (a) fresh human CLECs applied topically as a cell pellet layer with a density of 10^5^ cells/cm2, followed by coverage with standard wound dressings (CLEC); (b) cryopreserved human skin allografts (HSG) consented for research from the Singapore General Hospital Skin Bank [25] (Appendix A), anchored with 4/0 Vicryl sutures followed by coverage with standard dressings; and (c) standard wound dressing coverage alone (“Untreated”).

Configuration 2: Using a total of six pigs (separated into two sets of three pigs), each of the treatment arms mentioned in configuration 1 were used in its entirety on all six wounds per pig (Figure 4b). This was to eliminate the potential confounding effect to the systemic response on wound healing due to the multiple treatments on the same pig as used in configuration 1. In this configuration, one of the wounds was designated for skin punch biopsy samples to be collected for further histological analysis but would not be used in wound analysis. In Set A of configuration 2, the wounds were followed up for 7 weeks, while in Set B the pigs of the respective treatment arms were sacrificed once the average closure of all wounds on the pig achieved 90% or above.

In the two above-described configurations, all treatment arms were applied on the created wounds at Day 0. Analgesics (Carprofen 2–4 mg/kg) were given prior to all procedures that caused distress/pain and when necessary, based on the vet’s discretion. The standard wound dressings consisted of Jelonet (Smith & Nephew, Hull, UK) sterile paraffin tulle gras dressing as the first layer applied to the wound with a secondary overlay of bulky dry gauze dressings to provide some degree of bolstering and a final layer of Opsite (Smith & Nephew) transparent adhesive film dressing. All wound dressings were further secured by crepe bandages wrapped around the torso of the animal, secured with an outer animal jacket, and changed weekly during wound inspections.

### 4.6. Wound Healing Analysis

During the weekly wound dressing changes and inspections, the wounds were gently washed with saline to remove any debris before digital images of the wounds were taken. These images were analyzed with an open-source digital imaging software, MacBiophotonics ImageJ^TM^ (National Institutes of Health, Bethesda, MD, USA) for the rate of wound closure across all of the three treatment arms over the assessment period. Complete wound closure was defined as 100% re-epithelization without drainage. The equation (original wound area—new wound area)/original wound area × 100% was used to calculate percentage of wound closure. For Masson’s Trichome (MT) and immunohistochemistry (IHC) stainings, 5 mm punch skin biopsies were also taken weekly from a single designated wound of each pig, and these wounds were excluded for wound healing assessment.

### 4.7. Histological Staining and Analysis

Human organotypic cultures were embedded in OCT compound and snap-frozen in liquid nitrogen. The OCT blocks were sectioned into 7 μm sections onto a glass slide using CM1900 or CM1850 cryostat (Leica Biosystems, Wetzlar, Germany), stained with H&E on Autostainer XL (Leica Biosystems), according to standard protocol [30]. These sections were also immunostained to determine the presence of epithelial markers and adhesion molecules. Briefly, the sections were fixed in 4% paraformaldehyde in 1× PBS for 15 min and permeabilized in PBS/0.1% Triton X-100 solution for 5 min. Samples were washed three times with PBS, followed by blocking with 5% goat serum for 15 min and incubation with primary antibodies overnight at 4 °C. Primary antibodies and their used dilution factor are shown in Appendix A. Following washes with PBS, the samples were incubated with fluorescence-labeled secondary antibodies for 1 h at RT to visualize the antigens. Secondary antibodies used included Alexa-Fluor 647-conjugated goat anti-mouse (1:1000, Life Technologies), Alexa-Fluor 488-conjugated goat anti-mouse (1:1000, Life Technologies), and Alexa-Fluor 488-conjugated goat anti-rabbit (1:1000, Life Technologies). Thereafter, nuclei of the samples were counterstained and mounted with ProLong^TM^ Gold Antifade Reagent with DAPI (Life Technologies) and visualized under a Leica DMi8 fluorescent Microscope (Leica Biosystems).

Next, porcine skin punch biopsies were embedded in paraffin and sectioned into 4 μm slices onto glass slides using Microtome RM2235 (Leica Biosystems). Sections were processed according to the standard protocol for MT and IHC stainings. In the IHC stains, mouse anti-CD4 (1:50, Biorad, Hercules, CA, USA) and mouse anti-CD8 (1:100, Biorad) were used to stain the pig skin samples on Leica Bond III Automated Stainers (Leica Biosystems). The following were the default automated staining protocol (IHC DAB1 Leica Bond III): dewaxing at 72 °C, pre-treatment (unmasking) with Bond epitope retrieval solution 2 and wash steps using absolute alcohol and Bond Wash Solution, staining with primary antibodies, and detection using Bond™ Polymer Refine Detection. After staining, the slides were dehydrated in absolute alcohol, cleared with xylene, and finally mounted in depex. The stained slides were subsequently scanned using Ultra-Fast Scanner 1.6 (Philips, Amsterdam, The Netherlands).

Semi-quantitative analyses were respectively performed on the MT [15] and IHC [16] (blinded) stains by a pathologist.

### 4.8. Milliplex Cytokine Array

Porcine whole bloods were obtained weekly from pre-operation till Week 7 post excisional wound procedure. The bloods were centrifuged and the serum that was obtained was stored at −80 °C. Pre-operation serums were used as the baseline for the respective treatment groups. Anti-inflammatory and pro-inflammatory cytokines (GM-CSF, IFN-γ, IL-1α, IL-1β, IL-2, IL-4, IL-6, IL-8, IL-10, IL-12, IL-18, and TNFα) present in the porcine serum were detected through the use of Milliplex Porcine Cytokine/Chemokine Magnetic Bead Panel Kit (EMD Millipore Corporation, Burlington, MA, USA), performed in duplicates or triplicates according to the manufacturer’s protocol with Luminex Sheath Fluid (Luminex Corporation, Austin, TX, USA) and handheld magnetic separation block (EMD Millipore Corporation). The detection of cytokines levels was subsequently analyzed using Luminex 200 software (Luminex Corporation).

### 4.9. Statistical Analysis

Statistical analysis using one-way ANOVA with Bonferroni’s post-test was used for the comparison for the percentage of positively stained cells taken from three different donors for CLECs or KCs with two technical duplicates in flow cytometry analysis, as well as on tattooed wound contraction when the animals were sacrificed across the three treatment arms of CLEC, HSG, and standard dressings (no treatment). Two-way repeated measures ANOVA with Bonferroni’s post-test was used to analyze and to compare the percentage of wound closure on the weekly inspected porcine wounds managed by the three treatment arms. All statistical analyses were performed using GraphPad Prism software (version 5.01) for Windows (GraphPad Software Inc., San Diego, CA, USA).

## 5. Conclusions

In summary, while CLECs were previously purported to have common features of KCs at the morphological, molecular, and functional level [13], as well as having stem cell-like properties [20], we found significantly lower expression levels of *ITGB1* and *KRT15* in CLECs based on flow cytometry analysis. These two markers were previously associated with the basal and the epidermal stem cell characteristics of KCs [22,31]. We further confirmed the capability of CLECs to fully stratify in skin organotypic cultures even though the generated epithelium appeared less mature compared to the one formed by KCs. In addition, the location of basal/epidermal stem cell markers of *KRT14*, *KRT15*, and *p63* were found to be expressed at both the basal and the suprabasal layers in the CLEC-generated epithelium, unlike in the KC-generated epithelium which was restricted mainly to the basal layer. Despite these discrepancies with KCs, CLECs administered topically on full thickness skin wounds in the pig, was able to accelerate healing with no observed adverse reaction and overt immune rejection. While further in vivo work is needed to delineate the mechanisms behind the improved healing and transplantation immunology, this first-in-porcine study using CLECs demonstrates the safety and the potential of these cells as a non-native source to provide an epithelial barrier and promote wound healing.

## Figures and Tables

**Figure 1 ijms-23-08918-f001:**
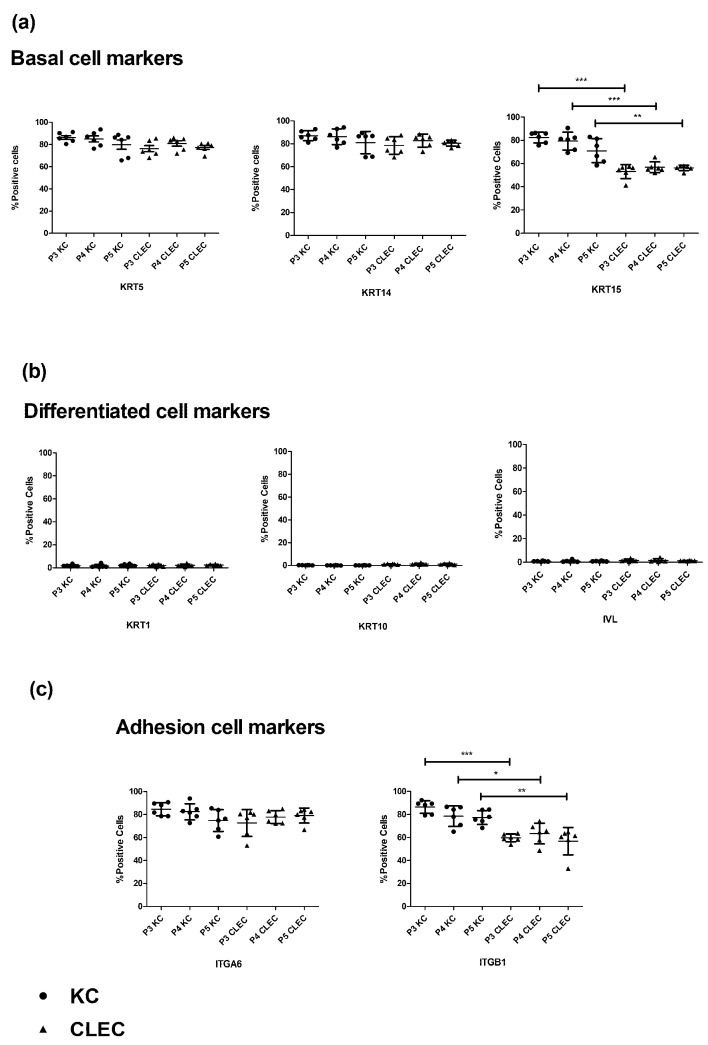
Dot plots based on flow cytometry analysis of passages three to five KCs and CLECs (three donors each cell type with two technical replicates), depicting the immunophenotypic expression levels of epidermal (**a**) basal cell markers using KRT5, KRT14, and KRT15; (**b**) differentiation cell markers using KRT1, KRT10, and IVL; and (**c**) adhesion cell markers using ITGA6 and ITGB1. Center line of the dot plots is the mean, and the whiskers represent standard deviation (SD). Statistical analysis was performed using one-way ANOVA—Bonferroni’s Multiple Comparison Test (n = 6); * *p* < 0.05, ** *p* < 0.001, and *** *p* < 0.0001.

**Figure 2 ijms-23-08918-f002:**
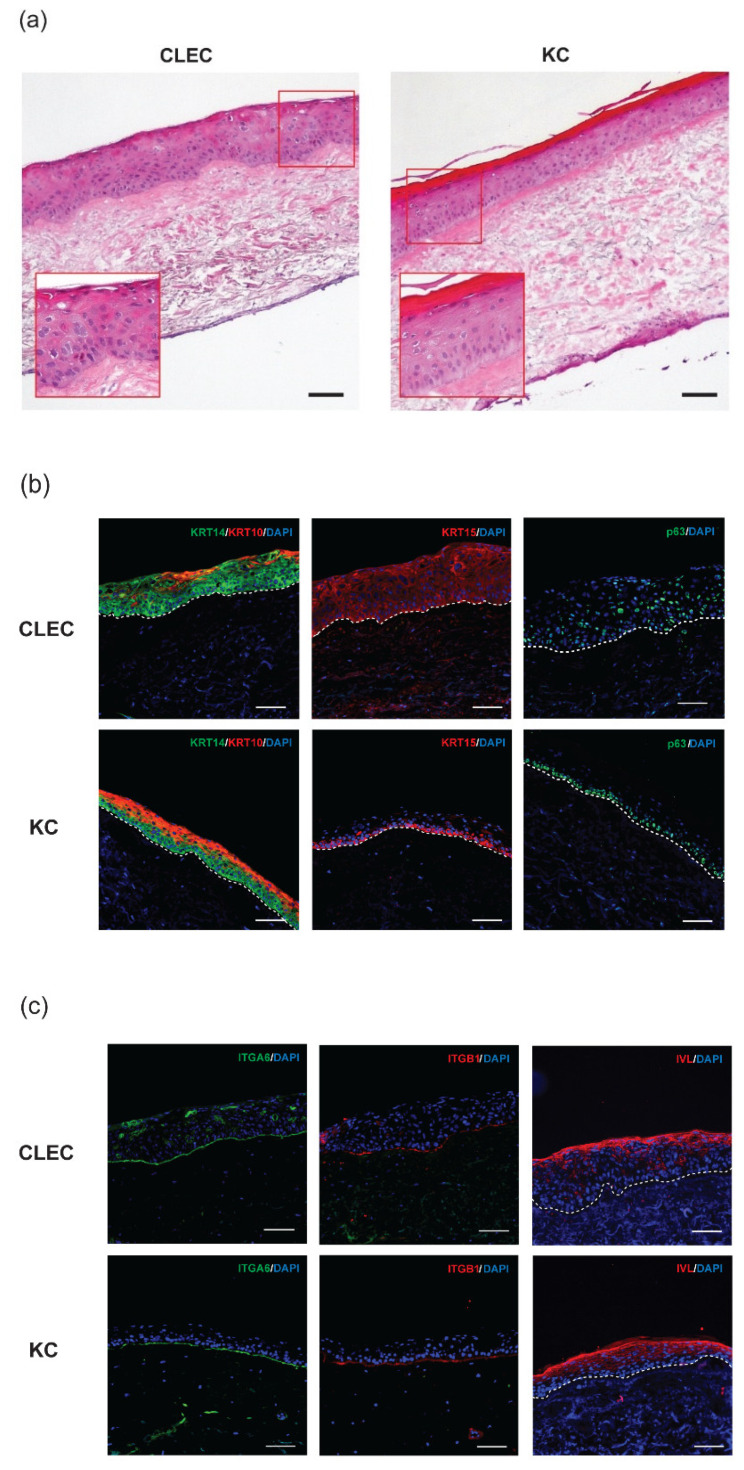
Epithelia formed by CLECs and KCs seeded on de-epithelialized dermis populated with human dermal fibroblasts. After 2 weeks of air-liquid interface exposure, these organotypic cultures were examined by (**a**) H&E stains; inset: higher magnification (20×) and IF stains of (**b**) KRT10, KRT14, KRT15, and p63; (**c**) ITGA6, ITGB1, and IVL. Scale bar = 100 µm.

**Figure 3 ijms-23-08918-f003:**
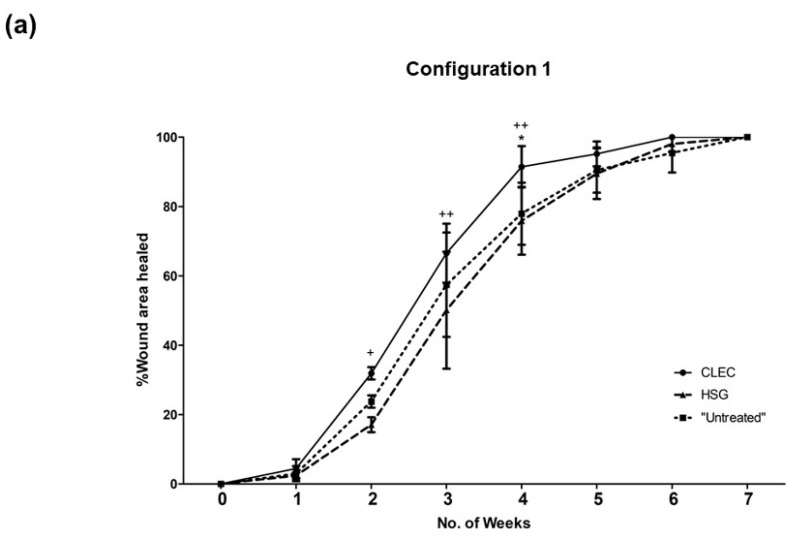
Percentage closure of weekly-measured wound area represented by mean ± SD based on (**a**) Configuration one with all the three treatment arms performed on a single pig (total pigs used: 2); (**b**) Configuration 2 Set A and (**c**) Configuration 2 Set B where only a single type of treatment was administered entirely to each pig (total pigs used: 6). The three treatment arms compared were CLEC, HSG, and standard dressings (“untreated”); n = 4 for configuration 1 and n = 5 for configuration 2. Statistical analysis was performed using one-way ANOVA—Bonferroni’s Multiple Comparison Test. Representative images of the stages of wound closure were shown for Configuration 2 at different time points with scale bar = 1 cm.

**Figure 4 ijms-23-08918-f004:**
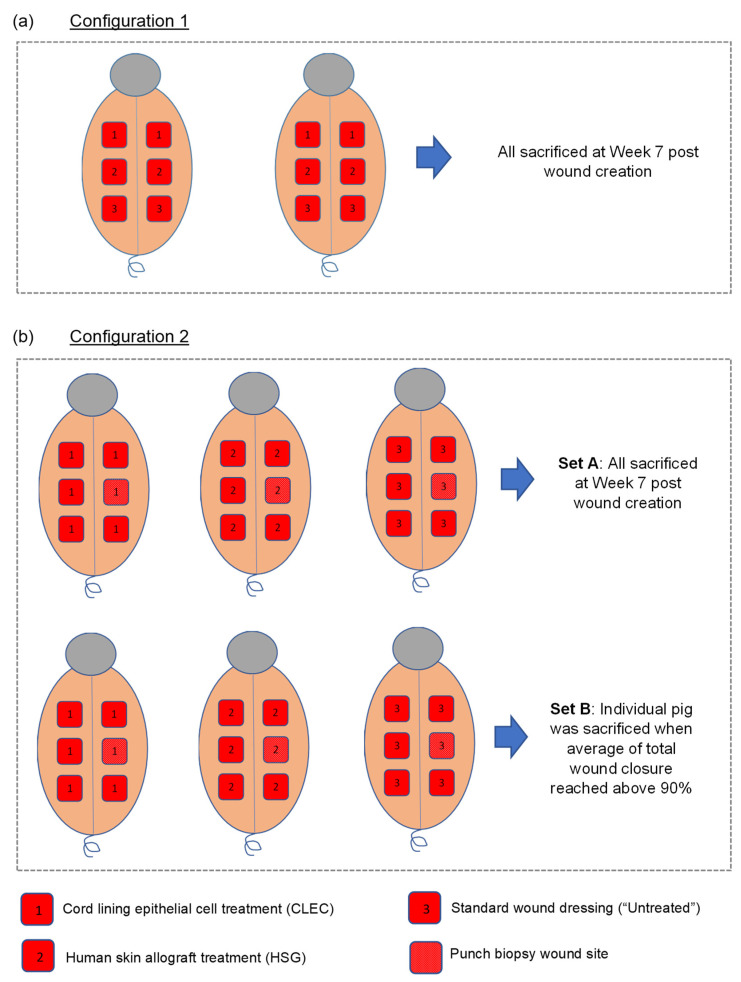
Wound treatment configurations for pigs created with full thickness excisional wounds. (**a**) Configuration 1 involved the use of multiple treatments on a single pig; (**b**) Configuration 2 involved the use of singular treatment on an entire pig.

**Figure 5 ijms-23-08918-f005:**
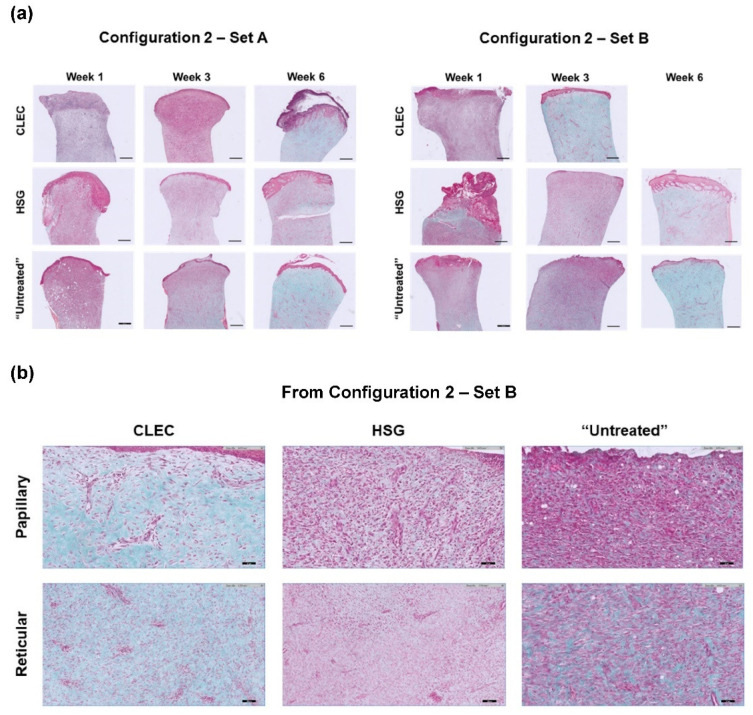
Week 3 tissue biopsies of pig cutaneous wounds treated with CLECs (CLEC), human skin grafts (HSG), and standard dressings (“Untreated”) probed with Masson’s Trichome (MT) stain (**a**) at 2× magnification for three time points in Configuration 2 Sets A and B with scale bar = 500 µm; (**b**) at 20× magnification based on Configuration 2 Set B with scale bar = 50 µm.

**Figure 6 ijms-23-08918-f006:**
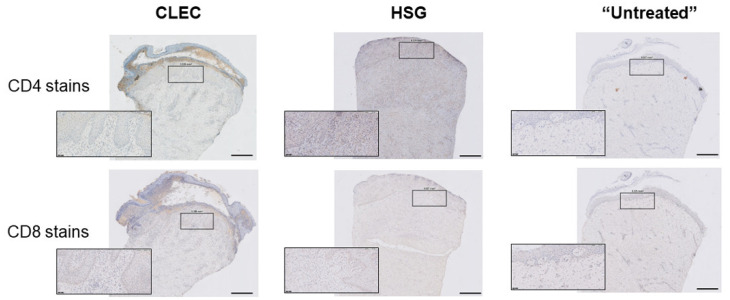
Immunohistochemical (IHC) stains of CD4 and CD8 on tissue biopsies from pig wounds treated with CLECs (CLEC), human skin grafts (HSG), and standard dressings (“Untreated”). Inset: higher magnification (20×). Main scale bar = 500 µm and inset scale bar = 50 µm.

**Table 1 ijms-23-08918-t001:** Location and distribution of markers associated with human skin epidermis, basement membrane, and epidermal stem cells within the entire epithelia reconstituted by KCs or CLECs in organotypic cultures.

Protein Marker	CLEC	KC
Keratin 10 (KRT10)	sb (+)	sb (++)
Keratin 14 (KRT14)	bl (++), sb (++)	bl (++)
Keratin 15 (KRT15)	bl (++), sb (++)	bl (++)
Integrin alpha-6 (ITGA6)	bm (++)	bm (++)
Integrin beta-1 (ITGB1)	bm (++)	bm (++)
Involucrin (IVL)	usb (++)	usb (++)
p63	bl (+), sb (+)	bm (++)

usb: upper suprabasal layer, sb: suprabasal layer, bl: basal layer, bm: present in basement membrane, +: present in parts of layer or region, ++: present everywhere in layer or region.

**Table 2 ijms-23-08918-t002:** Semi-quantitative scoring of Masson’s Trichome (MT) stains at the papillary and recticular dermis of skin tissue biopsies shown in Figure 5a.

	Configuration 2—Set A			Configuration 2—Set B		
Week	1	3	6	1	3	6
CLEC treatment						
Papillary	-	-	+	-	+	NA
Recticular	-	+	+	+	+	NA
HSG treatment						
Papillary	-	-	+	-	-	+
Recticular	-	-	+	-	-	+
“Untreated”						
Papillary	-	-	+	-	-	+
Recticular	-	+	+	+	+	+

“+”: Strong blue MT staining; “-“, Negative MT staining; “NA”: Not applicable.

**Table 3 ijms-23-08918-t003:** Distribution levels of CD4+- and CD8+-T lymphocytes infiltration into wound bed via representative tissue biopsies of each treated arm.

For CD4+ T cells								
				**Week**				
	**1**	**2**	**3**	**4**	**5**	**6**	**7**	**8**
CLEC treatment								
Set A	+++	+	+	++	-	+	+	NA
Set B	+++	+	-	-	NA	NA	NA	NA
HSG treatment								
Set A	+++	+	+	-	+	++	++	NA
Set B	++	++	+	++	++	++	-	-
“Untreated”								
Set A	+	++	+	+	++	-	+	NA
Set B	+++	+	+	+	-	+	NA	NA
For CD8+ T cells								
				**Week**				
	**1**	**2**	**3**	**4**	**5**	**6**	**7**	**8**
CLEC treatment								
Set A	++	++	++	+	+	+	+	NA
Set B	++	++	++	++	NA	NA	NA	NA
HSG treatment								
Set A	++	++	++	++	++	+	+	NA
Set B	+	++	++	++	++	++	++	+
“Untreated”								
Set A	+	++	++	++	++	++	++	NA
Set B	++	++	++	++	+	+	NA	NA

“-“: Absent, “+”: Minimal or sparsely distributed, “++”: Moderately distributed, “+++”: Severe or densely populated, “NA”: Not applicable.

**Table 4 ijms-23-08918-t004:** Fold change increase of probed inflammatory cytokine concentrations in porcine blood serum from the pre-procedure baseline level; only cytokines with a fold change of two and above are reflected in the table.

CLEC treatment								
	Day 4	Week 1	Week 2	Week 3	Week 4	Week 5	Week 6	Week 7
IFN-γ	+							
IL-8	++++				+++			
TNF-α	+							
HSG treatment								
	Day 4	Week 1	Week 2	Week 3	Week 4	Week 5	Week 6	Week 7
IFN-γ	+							
IL-1β			++				+	
IL-2			+					
IL-6			+					
IL-8	++++			+++	++++	++++	+++	
IL-10			+	+				
IL-12			+		+			
IL-18			+		+			
TNF-α				+++	+++	+++	++	
“Untreated”								
	Day 4	Week 1	Week 2	Week 3	Week 4	Week 5	Week 6	Week 7
IL-10				+		+		+

+: 2 to 10 fold increase, ++: 11 to 30 fold increase, +++: 31 to 60 fold increase, ++++: 61 to 120 fold increase.

## Data Availability

The raw data supporting the conclusions of this article will be made available by the authors, without undue reservation.

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
