# Peer review of "Human Umbilical Cord Lining-Derived Epithelial Cells: A Potential Source of Non-Native Epithelial Cells That Accelerate Healing in a Porcine Cutaneous Wound Model"

_ijms, 2022, doi:10.3390/ijms23168918_

Round 1

Reviewer 1 Report

This paper represents another study using cells derived from the umbilical cord. The stated novelty is that these cells are different from the 'other' cells isolated from the umbilical cord by other groups using different techniques such as isolation from wharton's jelly. Since this paper is focused on the clinical use of these cells I do not see enough evidence of the advantages of these cells over other cells. The authors compare CLECs to keratinicytes but they should be compared to other umbilical cord cells isolated and used by other groups.

The animal model is a great model for doing these direct comparisons since a number of wounds can be made and treated on the same individual. This eliminates some of the technical issues of using mice for example.

Although the study is a good, standard study, the data is weak. There does not seem to be a big advantage of using the cells over no treatment. Yes, the closure is 70% versus 100% of the wound and is statistically significant but this is still not a large enough differences to merit using cells versus current treatments when you consider the regulatory and economic hurdles of cell treatment versus medical bandages.

The result observed here is also observed in other studies that used bone marrow MSC,  Cord tissue MSC and cord blood cells. Some of the possible interesting advantages that might be observed would be with wound repair quality. A comparison by a pathologist of scarring versus regeneration with proper skin architecture would be helpful, if there was a difference seen. Fast wound healing that results in scar tissue is not as advantageous and slower, but proper healing. The authors hint as skin quality with their Masson Trichrome staining but do not show high enough resolution pictures to determine if there is scaring or not. They also measure the amount of Collagen but this has to put into context. Normal skin has collagen and scar tissue has collagen, but the histology/morphology of the organization of the collagen observed will tell much about the healing process.

In summary, in order to elevate this paper above the 1000's of other cell-wound healing papers already published the authors should provide data explaining why these cells are superior to other umbilical cord cells. A comparison, in their wound model, between CLECs and other cord tissue derived cells would help establish CLECs as the 'go to' cell.

Importantly, high magnification and high resolution pictures of the masson-trichrome slides with an in depth analysis by a pathologist discussing the status of the wound repair would add important data about the healing capacity of CLECs.

Author Response

Comments and Suggestions for Authors
This paper represents another study using cells derived from the umbilical cord. The stated novelty is
that these cells are different from the 'other' cells isolated from the umbilical cord by other groups
using different techniques such as isolation from wharton's jelly. Since this paper is focused on the
clinical use of these cells I do not see enough evidence of the advantages of these cells over other
cells. The authors compare CLECs to keratinicytes but they should be compared to other umbilical
cord cells isolated and used by other groups.

We thank this reviewer for the frank comment above and understandably so, as there are so many
papers out there using umbilical cord-derived derived cells, especially mesenchymal stromal cells to
improve wound healing and some of these are probably in clinical trials.

However, skin keratinocytes, whether they are for autologous or allogenic use, were really the
forerunner, as cultured cell therapy. These cells were utilised for wound healing in chronic wounds and
burns in the clinics, starting as early as in the 80s [1-3]; and were used even for permanent coverage
of large and severe burns [4]. On top of secreting the necessary cytokines and chemokines to stimulate
wound healing, keratinocytes/epithelial cells have the additional advantage of providing biological
barrier as an important function for wounds, something that mesenchymal stromal cells clearly cannot
provide. This barrier function is critical for protecting the wound bed from environmental pathogens and
preventing dehydration to the patients with large surface area wounds.

Therefore, we chose to study the in vitro and in vivo performance of CLECs compared to skin
keratinocytes and allografts which are now routinely used in many burns centre worldwide [5, 6].

The animal model is a great model for doing these direct comparisons since a number of wounds can
be made and treated on the same individual. This eliminates some of the technical issues of using
mice for example.

Although the study is a good, standard study, the data is weak. There does not seem to be a big
advantage of using the cells over no treatment. Yes, the closure is 70% versus 100% of the wound
and is statistically significant but this is still not a large enough differences to merit using cells versus
current treatments when you consider the regulatory and economic hurdles of cell treatment versus
medical bandages.

We appreciate the reviewer for recognizing the advantage of using the porcine model for our wound
study and, highlighting the concern of whether the improvement in wound healing seen in our animal
model is compelling enough to switch over to CLECs, considering the regulatory and economical
hurdles.

This study started with a view that CLECs can serve two purposes.

First, to be cultured as an autologous source (if these tissues/cells are banked, like cord blood banking)
for patients with little or unsuitable skin donor sites to harvest skin keratinocytes for treatment, as
mentioned in the main text. If cultured autologous CLECs can be proven to engraft and regenerate on
full thickness cutaneous wounds (a further study which we are currently performing) just like cultured
autologous skin keratinocytes, which are now typically used as part of the life-saving procedures for
critical extensive burns or complex wounds; this would (in our opinion) be a worthy direction to move
towards to. CLECs could be an important autologous alternative to save lives for very severe
burn/wound injuries. Moreover, there is no difference in regulatory and economic hurdles to autologous
cultured keratinocytes and autologous cultured CLECs, typically having to conform to GMP standards
in developed countries.

Second, CLECs can be cultured as an allogenic source from the large pool of medical waste to mitigate
the low supply of skin allografts in many East Asian countries due to lack of organ/tissue donors. Human
skin allografts are known to be the gold standard of biological wound dressings used on excised burns
that can provide barrier protection and prevent dehydration as a stop-gap measure before definitive

coverage by autografts. Because of the lack of human donors, some regions have resorted to porcine
skin use instead [7]. However, this constitutes a risk of zoonotic crossover and infection which is highly
not recommended in the light of SARS and current COVID-19 pandemic. Therefore, having a human
alternative to skin allograft in the form of skin epithelium substitute by CLECs, in our opinion, seems to
be worthwhile as these cells can provide the necessary barrier function, prevent dehydration of the
patient and most of all, give us the safety assurance. While there will be cost incurred due to regulatory
and economical hurdles to culture these allogeneic CLECs, we think that these barriers will be mitigated
due to the abundance of umbilical cords as medical waste (hence more cost effective), higher number
of cells at isolation (up to 6 billion cells), their cell naivety status having been isolated from an extra-
embyonic source, and good growth potential.

The result observed here is also observed in other studies that used bone marrow MSC, Cord tissue
MSC and cord blood cells. Some of the possible interesting advantages that might be observed would
be with wound repair quality. A comparison by a pathologist of scarring versus regeneration with
proper skin architecture would be helpful, if there was a difference seen. Fast wound healing that
results in scar tissue is not as advantageous and slower, but proper healing. The authors hint as skin
quality with their Masson Trichrome staining but do not show high enough resolution pictures to
determine if there is scaring or not. They also measure the amount of Collagen but this has to put into
context. Normal skin has collagen and scar tissue has collagen, but the histology/morphology of the
organization of the collagen observed will tell much about the healing process.

We thank the reviewer once again for raising these pertinent points to improve the paper.

In the revision, we have adopted a semi-quantitative scoring method to score the MT stains by our
pathologist (JPSY) as shown in Table 2 and have also included the higher magnification figures of the
MT stains in Figure 4b. This is followed by the pathologist’s description placed in Section 2.4
(highlighted in manuscript) as follows:

Further analysis of the MT stains at higher magnification (Figure 4b) using semi-quantitative scoring
method [15] (Table 2) revealed that CLEC-treated skin stained with MT demonstrated strong blue
staining indicating new collagen formation in the region of reticular dermis as early as at Week 3. In the
Set B experiment, the blue staining could be found not only in the region of reticular dermis but also in
the papillary dermis as early as at Week 3, whereas the HSG-treated skin as well as the untreated skin
could only demonstrated the similar positive staining pattern at Week 6.

In addition, we have further included in Supplementary Figure 3 the analysis of wound contraction at
the point of sacrifice, from the tattoos created to mark out the initial boundary of the wound area created.
We found no difference in wound contraction across all the different treatment arms on the pigs. These
levels of wound contraction were consistent with an earlier report by Gallant-Belm & Hart [8] which
reported that healing of excisional wounds in Yorkshire pigs closely resembled normal healing in
humans with neither having excessive contraction nor forming hypercontracted scars.

In summary, in order to elevate this paper above the 1000's of other cell-wound healing papers
already published the authors should provide data explaining why these cells are superior to other
umbilical cord cells. A comparison, in their wound model, between CLECs and other cord tissue
derived cells would help establish CLECs as the 'go to' cell.

Importantly, high magnification and high resolution pictures of the masson-trichrome slides with an in
depth analysis by a pathologist discussing the status of the wound repair would add important data
about the healing capacity of CLECs.

We hope we have sufficiently addressed this reviewer’s concerns and in future work, we will look into
comparing CLECs with other umbilical cord cells as suggested. Thank you.

References:
1. Hefton JM, Madden MR, Finkelstein JL, Shires GT: Grafting of burn patients with allografts
of cultured epidermal cells. Lancet (London, England) 1983, 2(8347):428-430.

2. Leigh IM, Purkis PE: Culture grafted leg ulcers. Clin Exp Dermatol 1986, 11(6):650-652.

3. De Luca M, Albanese E, Bondanza S, Megna M, Ugozzoli L, Molina F, Cancedda R, Santi PL,
Bormioli M, Stella M et al: Multicentre experience in the treatment of burns with
autologous and allogenic cultured epithelium, fresh or preserved in a frozen state. Burns :
journal of the International Society for Burn Injuries 1989, 15(5):303-309.

4. Gallico GG, 3rd, O'Connor NE, Compton CC, Kehinde O, Green H: Permanent coverage of
large burn wounds with autologous cultured human epithelium. The New England journal
of medicine 1984, 311(7):448-451.

5. Wang C, Zhang F, Lineaweaver WC: Clinical Applications of Allograft Skin in Burn Care.
Annals of plastic surgery 2020, 84(3S Suppl 2):S158-s160.

6. Chua AW, Khoo YC, Tan BK, Tan KC, Foo CL, Chong SJ: Skin tissue engineering advances in
severe burns: review and therapeutic applications. Burns & trauma 2016, 4:3.

7. Chiu T, Burd A: "Xenograft" dressing in the treatment of burns. Clinics in dermatology 2005,
23(4):419-423.

8. Gallant-Behm CL, Hart DA: Genetic analysis of skin wound healing and scarring in a porcine
model. Wound repair and regeneration : official publication of the Wound Healing Society
[and] the European Tissue Repair Society 2006, 14(1):46-54

Reviewer 2 Report

The research work presented by Kua et al. is exciting, the research design is appropriate and the results are presented in a quite comprehensible way. The findings are quite interesting and the manuscript is overall of high scientific value. There are only a few points that need to be pointed out:

1. The authors show that basal markers of the human epidermis were found to be expressed in both the basal and suprabasal layers of CLEC-generated epithelium, with the overall appearance of CLEC-generated epithelium being less mature and organized compared to the one formed by KCs. The authors hypothesize correctly that this could be due to the naive nature of CLECs and that the phenomenon could alter later. Did the authors think to test this hypothesis by assessing basal markers in later time points in vitro? Could this also be applicable to skin samples from the animals eg at week 6?

2. The authors have assessed infiltration by CD4+ and CD8+ cells. CD8+ cells seem to be moderately present in all weeks of the experiments, however, the authors have not commented on this result.

3. Please define the composition or provide an identification code for the PTT-e3 medium.

3. Please check some language/structure inconsistencies e.g.:

lines 56-59, the sentence "On top of the advantages.." needs re-arrangement.

lines 204-207, there is something missing from the sentence.

line 466, the word "immune".

Author Response

Comments and Suggestions for Authors
The research work presented by Kua et al. is exciting, the research design is appropriate and the
results are presented in a quite comprehensible way. The findings are quite interesting and the
manuscript is overall of high scientific value. There are only a few points that need to be pointed out:

We much appreciate the reviewer for the positive comments above which are highly encouraging to
the study team.

1. The authors show that basal markers of the human epidermis were found to be expressed in both
the basal and suprabasal layers of CLEC-generated epithelium, with the overall appearance of CLEC-
generated epithelium being less mature and organized compared to the one formed by KCs. The
authors hypothesize correctly that this could be due to the naive nature of CLECs and that the
phenomenon could alter later. Did the authors think to test this hypothesis by assessing basal
markers in later time points in vitro? Could this also be applicable to skin samples from the animals eg
at week 6?

We thank the reviewer for raising the above point about the basal marker expression on CLEC-
generated epithelium.

We have not yet tested the hypothesis for further prolonged stratification in air-liquid interface condition
in vitro. However, we are in the midst of preparing this experiment along with an in vivo engraftment
assay on nude mice which we hope to report in the future. We have an interest to translate the use of
CLECs eventually and understanding this aspect is important.

As for the question if the skin samples from the animals (say at week 6) is applicable, we do not think
so. This is because the human CLECs used on the pig’s wound bed are considered xenografts on an
immunocompetent animal and therefore with complications arising from immune response of the host,
there is no assurance that proper stratification of CLECs would take place on the wound bed. And that
is why, the in vivo engraftment assay on immunodeficient nude mice mentioned earlier is important for
us to answer this pertinent question of yours.

2. The authors have assessed infiltration by CD4+ and CD8+ cells. CD8+ cells seem to be moderately
present in all weeks of the experiments, however, the authors have not commented on this result.

We thank the reviewer for asking yet another pertinent question.

A 2014 time-course study of lymphocytic infiltration into 3mm diameter full thickness excisional skin
wounds of BALB/c mice by Chen et al. [1], found that CD4+ and CD8+ T cells infiltrate progressively
into the wounds (starting at Day 1 and Day 3 respectively), and remained present up to Day 21 when
the wounds were almost or completely re-epithelialized. Our current pig study similarly demonstrated
that CD4+ and CD8+ cells were present in the untreated wounds from Week 1 in our time course study
up to early wound closure.

As we are not aware of any other prior CD8+ T cells time-course tracking on pig wounds that was
reported, our porcine study does correlate in a way to the 2014 mice study above, even though
distribution of the CD8+ T cells present at the various time points differed. However, we are not able to
fully explain the lack of change in the CD8+ T cell distribution profile after the use of human CLECs and
skin grafts which are xenogeneic in nature. In these 2 cases, we can only postulate the CD8+ T cells
recruited into the wounds have reached a threshold regardless the trigger that was due to the wounding
itself, the immunogenic response to the foreign cells/tissues or a combination of both.

We have also put in the above explanation (shorter form) in Section 3.4 of the manuscript (highlighted
in the manuscript).

3. Please define the composition or provide an identification code for the PTT-e3 medium.
We have provided the medium composition in Section 4.1 of the manuscript (highlighted in the
manuscript).

4. Please check some language/structure inconsistencies e.g.:

lines 56-59, the sentence "On top of the advantages.." needs re-arrangement.

lines 204-207, there is something missing from the sentence.

line 466, the word "immune".

Our sincere thanks to the reviewer once again for watching out for us on these typo and inconsistencies
due to human error. We have amended them accordingly and have also done so for a few others which
we have spotted along the way.

Reference

1. Chen L, Mehta ND, Zhao Y, DiPietro LA: Absence of CD4 or CD8 lymphocytes changes
infiltration of inflammatory cells and profiles of cytokine expression in skin wounds, but
does not impair healing. Exp Dermatol 2014, 23(3):189-194